



**The response of diazotrophs to nutrient amendment in the**
**South China Sea and western North Pacific**
Zuozhu Wen[1,2], Thomas J. Browning[2], Rongbo Dai[1], Wenwei Wu[1], Weiying Li[1,a],
Xiaohua Hu[1], Wenfang Lin[1], Lifang Wang[1], Xin Liu[1], Zhimian Cao[1], Haizheng Hong[1],
and Dalin Shi[1]
[1]State Key Laboratory of Marine Environmental Science, Xiamen University, Xiamen,
Fujian, PR China
[2]Marine Biogeochemistry Division, GEOMAR Helmholtz Centre for Ocean Research
Kiel, Germany
*Correspondence to*: Dalin Shi (dshi@xmu.edu.cn) and Haizheng Hong
(honghz@xmu.edu.cn)
a. Present address: Key Laboratory of Marine Ecosystem Dynamics, Second Institute of
Oceanography, Ministry of Natural Resources, Hangzhou, Zhejiang, PR China





**Abstract.** The availability of iron (Fe) and phosphorus (P) have been shown to be key
factors regulating rates of nitrogen fixation in the western Subtropical Pacific. However,
their relative importance at finer spatial scales between the northern South China Sea
(NSCS) and the western boundary of the North Pacific is poorly constrained.
Furthermore, nutrient limitation of specific diazotroph types has not yet been assessed.
Here we investigated these unknowns by carrying out measurements of (i) finer scale
spatial variabilities in $N_2$ fixation rates and diazotroph abundances throughout these
regions, and (ii) conducting eight additional Fe and phosphate addition bioassay
experiments where both changes in $N_2$ fixation rates and the abundances of specific
diazotrophs were measured. Overall, nitrogen fixation rates were lower in the NSCS than
around the Luzon Strait and the western North Pacific, which we hypothesize was due to
lower Fe-to-fixed nitrogen supply ratios that decrease their competitive ability with non-
diazotrophic phytoplankton. The nutrient addition bioassay experiments demonstrated
that nitrogen fixation rates in the central northern South China Sea (NSCS) were co-
limited by Fe and P, whereas in the western boundary of the North Pacific they were P-
limited. Changes in the abundances of *nifH* in response to nutrient addition varied in how
well they correlated with changes in nitrogen fixation rates, and the largest responses
were always dominated by either *Trichodesmium* or UCYN-B. In general, nutrient
addition had a relatively restricted impact on diazotroph community structure apart from
on UCYN-B, which showed increased contribution to the diazotroph community



following P addition at sites where $N_2$ fixation rates were P-limited. We further
hypothesize the importance of absolute Fe supply rates in regulating spatial variability in
diazotroph community structure across the study area.



## 1 Introduction

Nitrogen fixation by diazotrophic bacteria converts abundant dinitrogen ($N_2$) gas into
ammonia, providing nearly half of the ocean's bioavailable nitrogen (N) (Gruber and
Galloway, 2008), which goes on to support >30% of carbon export from surface to deep
waters in the N-limited ocean (Böttjer et al., 2016; Wang et al., 2019). A diverse
community of diazotrophs has been described across the oligotrophic ocean that includes
*Trichodesmium*, unicellular cyanobacteria (UCYN-A and *Crocosphaera*, also referred to
as UCYN-B), the heterocystous symbiont *Richelia* associated with diatoms (DDAs,
diatom-diazotroph associations), and noncyanobacterial diazotrophs (NCDs,
heterotrophic or photoheterotrophic bacteria) (Zehr and Capone, 2020). However, there is
still a lack of knowledge on what controls diazotrophic distribution, activity and
community structure in the current ocean.

Iron (Fe) and phosphorus (P) are believed to be key factors controlling the biogeographic
distribution of marine $N_2$ fixation (Sohm et al., 2011; Zehr and Capone, 2020; Wen et al.,
2022). Fe is particularly important for $N_2$ fixers as a cofactor for the FeS-rich
nitrogenanse enzyme (Berman-Frank et al., 2001), whereas P is also required for genetic
information storage, cellular structure and energy generation. A number of nutrient-
addition bioassay experiments conducted in the field have shown that $N_2$ fixation in the
oligotrophic oceans can be limited by Fe or P, or co-limited by both nutrients at the same



time (Mills et al., 2004; Needoba et al., 2007; Grabowski et al., 2008; Watkins-Brandt et
al., 2011; Langlois et al., 2012; Dekaezemacker et al., 2013; Krupke et al., 2015; Tanita et
al., 2021; Wen et al., 2022; Turk-Kubo et al., 2012). However, few studies have
quantified how the supply of Fe and/or P impacts the abundance of individual
diazotrophic phylotypes and their community structure (Langlois et al., 2012; Moisander
et al., 2012; Turk-Kubo et al., 2012). Experiments conducted so far that investigated this
were located in the South Pacific and North Atlantic, and found diverse responses among
diazotrophic phylotypes to the addition of Fe and/or P. Furthermore, the responses of total
diazotroph abundances assessed from *nifH* gene quantifications were not qualitatively
match the responses of bulk $N_2$ fixation rates (Langlois et al., 2012; Moisander et al.,
2012; Turk-Kubo et al., 2012). Resolution of the specific types of diazotrophs responding
to nutrient supply, in addition to overall $N_2$ fixation rates, are potentially crucial for
understanding their biogeography, which in turn could be important for biogeochemical
function. For example, the presence of large *Trichodesmium* filaments is expected to have
a different fate in the microbial food web and contribute differently to the sinking flux of
carbon than that of small unicellular species (Bonnet et al., 2016).

The northern South China Sea (NSCS) and the neighboring western boundary of the
North Pacific are interacting water bodies, with the major western boundary Kuroshio
Current intruding into the NSCS across the Luzon Strait, generating frontal zones with



unique physical and biogeochemical characteristics (Du et al., 2013; Guo et al., 2017;
Huang et al., 2019; Li et al., 2021; Lu et al., 2019; Xu et al., 2018). Common to the full
regime, however, is surface waters that are warm, stratified and N-depleted, but subject to
elevated dust input from the Gobi Desert (Duce et al., 1991; Jickells et al., 2005). These
conditions potentially provide an ideal habitat for diazotrophs (Chen et al., 2003; Wu et
al., 2003). Investigations in these regions have shown high variability in diazotroph
abundances and $N_2$ fixation rates (Chen et al., 2003; Chen et al., 2014; Chen et al., 2008;
Lu et al., 2019; Wu et al., 2018), which overall increased from the NSCS basin to the
western boundary of the North Pacific (Wen et al., 2022). Along this gradient in $N_2$
fixation, the dominant diazotroph types switched from *Trichodesmium* in the NSCS to
UCYN-B in the western boundary of the North Pacific (Wen et al., 2022). Several studies
have hypothesized that these gradients of diazotroph abundances and $N_2$ fixation rates
were regulated by nutrient availability (specifically, Fe, P and N; Wu et al., 2003; Chen et
al., 2003; Chen et al., 2008; Shiozaki et al., 2014a; Shiozaki et al., 2015a). More recent
observational and experimental evidence supported the hypothesis that Fe:N supply ratios
are the main drivers of the abundance of diazotrophs and $N_2$ fixation rates across the
western North Pacific (Wen et al., 2022). With an increasing supply ratio of Fe:N from
the North Equatorial Current (NEC) to the Philippines Sea, Wen et al. (2022) found that
diazotroph abundances and $N_2$ fixation rates increased, and bioassay experiments
demonstrated evidence for $N_2$ fixation rates switching from Fe to P limitation or to



nutrient-replete conditions. In the NSCS, Wen et al. (2022) found $N_2$ fixation rates fell in
between NEC and Kuroshio values and bioassay experiments demonstrated rates were
co-limited by Fe and P, which they hypothesized was due to intermediate Fe:N supply
ratios (Wen et al., 2022).

Although this previous study has outlined the broad spatial pattern of nutrient regulation
of marine $N_2$ fixation throughout the western Subtropical Pacific (Wen et al., 2022),
important questions remain. Two specific examples are: (i) the relatively lower spatial
resolution of the experiments in Wen et al. (2022) and other studies (Chen et al., 2019;
Shiozaki et al., 2014b) remain insufficient to delineate Fe and P controls at finer spatial
scales between the neighboring NSCS and the western boundary of the North Pacific; and
(ii) In addition to controls on $N_2$ fixation rates, broad-scale differences in the types of
diazotrophs dominating the $N_2$ fixer community were not concretely associated with
environmental drivers in experimental tests for nutrient limitation, because changes in
type-specific diazotroph abundances following nutrient addition were not measured
(Chen et al., 2019; Shiozaki et al., 2014b; Wen et al., 2022). Therefore, in the present
study we extend the findings of Wen et al. (2022) and others by carrying out additional,
higher-spatial resolution observations of volumetric $N_2$ fixation rates and measurements
of the abundances of key diazotrophic phylotypes from the NSCS basin to the western
boundary of the North Pacific (including the upstream Kuroshio) between 2016 and 2018





(Fig. 1). These new observations were supplemented by a further additional eight, high
volume (10 L) nutrient amendment bioassay experiments throughout the transect to
directly test the response of both (i) $N_2$ fixation rates, and (ii) *nifH* gene abundances to
supply of potentially limiting nutrients (Fe, P, and Fe+P).

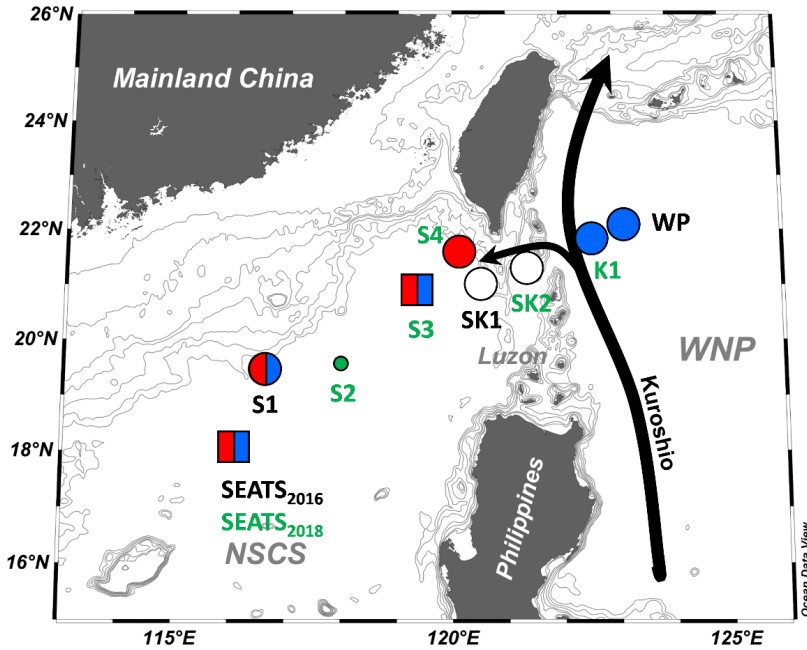

**Figure 1.** Sampling and nutrient amendment experiment locations in the northern South
China Sea and the western boundary of the North Pacific. One station (SEATS$_{2016}$) was
sampled in 2016, three (S1, SK1, WP) were in 2017, and six (stations with green labels)
were in 2018. Nutrient amendment experiments were conducted at 8 of 10 stations.
Symbols summarize the nutrient limitation of $N_2$ fixation rates found at each site: red, Fe
limitation; blue, P limitation; split red/blue, Fe-P co-limitation; white, nutrient replete.
Co-limitation type is indicated by symbol type (square, independent co-limitation; circle,
simultaneous co-limitation). WNP, the western North Pacific. Black arrows indicate
Kuroshio Current and its branch. Gray lines indicate 50, 100, 300, 500, 1000, 1500 and
2000 m bathymetric depth contours.




## 2 Method

### 2.1 Sample collection

Investigations and bioassay experiments were conducted on three cruises to the NSCS
(stations SETAS and S1 to S4), the Luzon Strait (stations SK1 and SK2), the upstream
Kuroshio (station K1), and the western boundary of the North Pacific (station WP) (Fig.
1), between May 2016 and June 2018 onboard the R/V *Dongfanghong* 2 and R/V *Tan*
*Kah Kee*. At each station (except station SK2 where no hydrological data are available),
temperature and salinity were recorded by a Seabird 911 CTD. Water samples were
collected using Niskin-X bottles at five or six depths (except SK2, only surface waters
were sampled) throughout the upper 150 m for the determination of $N_2$ fixation and
primary production rates. Seawaters from each depth were also sampled for the analysis
of *nifH* gene abundance. Samples for nutrient analysis were also collected. Seawater for
the bioassay experiments (at 8 of 10 stations) was collected using a trace-metal-clean
towed sampling device located around 2-5 m depth with suction provided by a Teflon
bellows pump. Seawaters were sampled in a dedicated trace-metal-clean laminar flow
hood maintained over-pressurized by HEPA-filtered air. During the cruise in 2018
(stations with green labels in Fig. 1), surface waters were sampled under trace-metal-
clean condition for the determination of total particulate Fe concentration.




## 2.2 N$_2$ fixation and primary production rate measurements

N$_2$ fixation rates were determined by the $^{15}$N$_2$ gas dissolution method (Mohr et al., 2010),

combined with a primary production assay using NaH$^{13}$CO$_3$ (99 atom% $^{13}$C, Cambridge

Isotope Laboratories). Briefly, 0.22 µm-filtered surface seawater was degassed using a

Sterapore membrane unit (20M1500A: Mitsubishi Rayon Co., Ltd., Tokyo, Japan) as

described in Shiozaki et al. (2015b). After that, 20 mL 98.9 atom% pure $^{15}$N$_2$ gas

(Cambridge Isotope Laboratories) was injected into a gas-tight plastic bag containing 2 L

of the degassed seawater and allowed to fully equilibrate before use. The N$_2$ fixation and

primary production incubations were conducted in duplicate 4.3 L Nalgene polycarbonate

bottles. Samples were spiked with 100 mL $^{15}$N$_2$ enriched filtered seawater from the same

site and incubated on-deck for 24 h. The final $^{15}$N$_2$ enriched seawater concentration in the

incubation bottles was not measured directly during this study. We thus employed a $^{15}$N$_2$

atom% of 1.40 ± 0.08 atom% ($n$ = 17) measured in a following cruise in 2020 (Wen et al.,

2022), during which the N$_2$ fixation incubations were conducted using the same

approach, reagents, and equipment as for the study described here. For primary

production measurements, NaH$^{13}$CO$_3$ solution was added at a concentration of 100 µM.

After that, the bottles were covered with a neutral-density screen to adjust the light to the

levels at sampling depths, and then were incubated for 24 h in an on-deck incubator

continuously flushed with surface seawater. Incubated samples were filtered onto pre-

combusted (450 °C, 4 h) GF/F filters, and the particulate organic matter from each depth



were also collected to determine background POC/PON concentrations and their natural
$^{13}$C/$^{15}$N abundances.

All filter samples were acid fumed to remove the inorganic carbon and then analyzed
using an elemental analyzer coupled to a mass spectrometer (EA-IRMS, Thermo Fisher
Flash HT 2000-Delta V plus). The $N_2$ fixation and primary production rates were then
calculated according to Montoya et al. (1996) and Hama et al. (1983), respectively. The
detection limits of $N_2$ fixation rates were then calculated according to Montoya et al.
(1996), taking 4‰ as the minimum acceptable change in the $\delta^{15}$N of particulate nitrogen.
All parameters involved in $N_2$ fixation rate calculation are shown in Supplementary
Materials. To represent the inventories, the upper 150 m depth-integrated $N_2$ fixation rate
and primary production were calculated by the trapezoidal integration method.

**2.3 *nifH* gene abundance**
At each depth, 4.3 L seawater samples for DNA extraction were filtered onto 0.22 μm
pore-sized membrane filters (Supor200, Pall Gelman, NY, USA) and then frozen in liquid
$N_2$. To extract the DNA, membranes were cut into pieces under sterile conditions, and
then extracted using the QIAamp® DNA Mini Kit (Qiagen) following the manufacturer's
protocol. The quantitative polymerase chain reaction (qPCR) analysis was targeted on the
*nifH* phylotypes of *Trichodesmium* spp., unicellular cyanobacterial UCYN-A1, UCYN-



A2, and UCYN-B, *Richelia* spp. (het-1), and a gamma-proteobacterium (γ-24774A11),
using previously designed primers and probe sets (Supplementary Table S1; Church et al.,
2005a; Church et al., 2005b; Moisander et al., 2008; Thompson et al., 2014). A recent
study suggested that the primers for UCYN-A2 also target UCYN-A3 and thus cannot be
used to differentiate between these two phylotypes (Farnelid et al., 2016). Therefore, we
used the convention UCYN-A2/A3 when referring to these two groups. The *nifH*
standards were obtained by cloning the environmental sequences from previous samples
collected from the SCS. qPCR analysis was carried out as described previously (Church
et al., 2005a) with slight modifications. Triplicate qPCR reactions were run for each
environmental DNA sample and for each standard on a CFX96 Real-Time System (Bio-
Rad Laboratories). Standards corresponding to between $10^1$ and $10^7$ copies per well were
amplified in the same 96-well plate. The amplification efficiencies of PCR were always
between 90-105%, with $R^2$ values > 0.99. The quantification limit of the qPCR reactions
was 10 *nifH* gene copies per reaction, and 1 μL from 100 or 150 μL template DNA was
applied to qPCR assay, which was equivalent to approximately ~230-350 gene copies per
L of seawater sample filtered (4.3 L).

**2.4 Bioassay experiments**
Acid-cleaned Nalgene polycarbonate carboys (10 L) were filled with near surface
seawater from the towed fish system. Trace metal clean techniques were strictly applied





in experimental setup and manipulations. All materials coming in contact with the
incubation water were acid-washed in a Class-100 cleanroom before use. Nutrient
amendments at all sites were Fe, P, and Fe+P. The amended Fe and P (chelexed and filter-
sterilized) concentrations were 2 nM and 100 nM, respectively. Control bottles incubated
with no nutrient treatment were included in all experiments. All treatments were
conducted with 2 or 3 replicates and incubated for 3 days in a screened on-deck incubator
continuously flushed with surface seawater. After pre-incubation, subsamples were
collected for the determination of $N_2$ fixation rate and *nifH* gene abundance. $^{15}N_2$
enriched seawater was prepared as described above, except that all the materials coming
in contact with the seawater were acid-cleaned before use.

**2.5 Macronutrient and chlorophyll *a* analyses**
Samples for macronutrient analyses were collected in 125-mL acid-washed high-density
polyethylene (HDPE) bottles (Nalgene), and analyzed onboard using a Four-channel
Continuous Flow Technicon AA3 Auto-Analyzer (Bran-Lube GmbH). The detection
limits for $NO_3^-+NO_2^-$ and $PO_4^{3-}$ were 0.1 μmol L$^{-1}$ and 0.08 μmol L$^{-1}$, respectively. The
nitracline was defined as the depth at which $NO_X$ concentration equaled 0.1 μmol L$^{-1}$ (Le
Borgne et al., 2002). Samples for chlorophyll *a* analysis were collected on nominal 0.7
μm pore-size GF/F filters (Whatman) and chlorophyll *a* concentration was determined
using a Trilogy fluorometer (Turner-Designs, USA).




### 2.6 Particulate Fe concentration

Total particulate Fe ($PFe_{total}$) and intracellular Fe ($PFe_{intra}$) were sampled under laminar
flow hood. Briefly, 4-9 L of surface waters were filtered onto acid-cleaned 0.22-μm
polycarbonate membrane filters. For $PFe_{intra}$ samples, in order to remove metal-bound to
the cell surface, cells were exposed twice to an oxalate-EDTA solution for 5 minutes and
rinsed nine times with Chelex-cleaned 0.56 mol $L^{-1}$ NaCl solution (Li et al., 2020).
$PFe_{total}$ and $PFe_{intra}$ concentrations were then determined by ICP-MS (ICP-MS 7700X,
Agilent).

### 2.7 Statistical analysis

Significance of differences among nutrient treatments of bioassay experiments (for $N_2$
fixation rate) were tested by ANOVA followed by Fisher PSLD test, using R-4.1.2.
Pairwise correlation between $N_2$ fixation rates, diazotroph groups and environmental
factors was analyzed using Pearson correlation. A significance level of $p < 0.05$ was
applied, except as noted where significance was even greater.

## 3 Results

### 3.1 Spatial variations of $N_2$ fixation rates and diazotroph composition

Our survey revealed substantial spatial variability in $N_2$ fixation rates and *nifH* gene
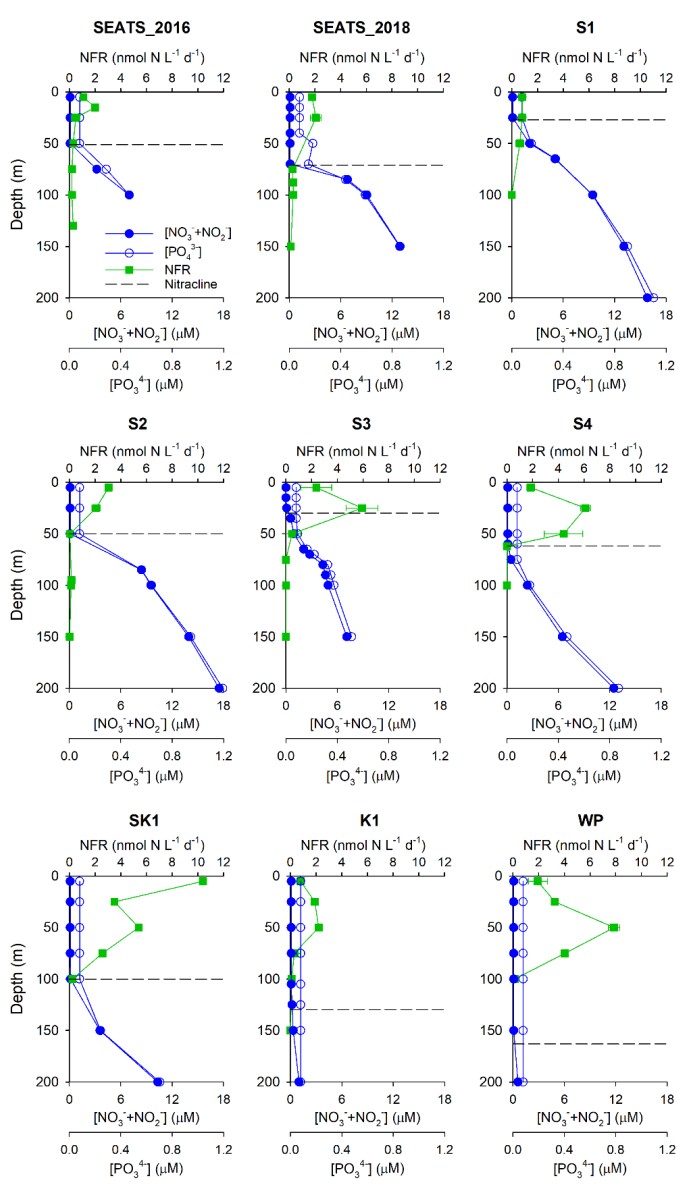

**Figure 2.** Vertical profiles of $N_2$ fixation rates. Green squares, $N_2$ fixation rate (NFR, nmol N $L^{-1}$ $d^{-1}$); blue solid circles, $NO_3^-$+$NO_2^-$ concentrations (µM); blue open circles, $PO_4^{3-}$ concentrations (µM). The dashed line indicates the nitracline depth. Note that no profile data were available at station SK2.



abundances across the study area (Figs. 2 and 3). Vertically, high $N_2$ fixation rates were
found in the upper 50 m (ranged from bellow detection limit to $10.4 \pm 0.01$ nmol N $L^{-1}$ $d^{-1}$),
rates dropped rapidly at greater depths (Fig. 2), and surface rates were positively
correlated with depth-integrated rates (Pearson $r = 0.68$, $p = 0.043$, Supplementary Table
S2). Horizontally, depth-integrated $N_2$ fixation rates were generally low at the central
NSCS basin stations (SEATS, S1 and S2, on average $86 \pm 33$ µmol N $m^{-2}$ $d^{-1}$), elevated at
stations close to the western edge of the Luzon Strait (S3 and S4, on average $214 \pm 47$ µmol
N $m^{-2}$ $d^{-1}$), and were highest at the Luzon Strait station (SK1, 437 µmol N $m^{-2}$ $d^{-1}$) and the
western North Pacific boundary station (WP, 403 µmol N $m^{-2}$ $d^{-1}$) (Figs. 1, 3 and Table 1).

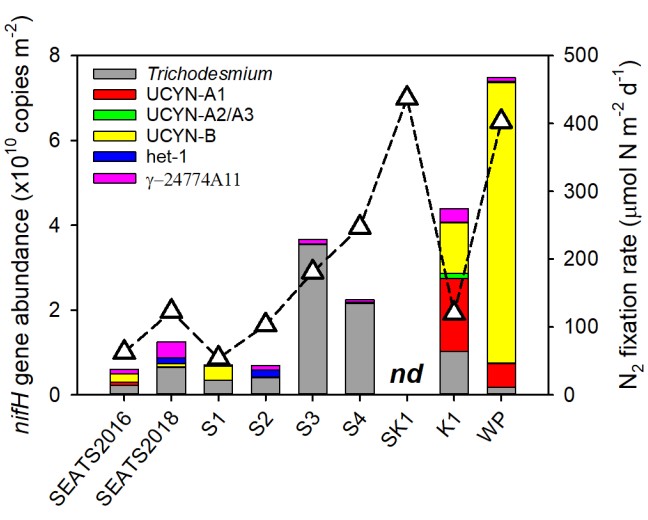



**Figure 3.** Depth-integrated (upper 150 m) *nifH* gene abundances (bars) and $N_2$ fixation
rates (triangles). Note that depth-integrated $N_2$ fixation rates and *nifH* gene abundances
were not available at station SK2. nd, not determined.




**Table 1.** Environmental conditions, $N_2$ fixation, and primary production rates. Sea
surface temperature (SST) and salinity (SSS), nitracline depth ($D_{Nitr}$), surface $N_2$ fixation
rate (SNF), upper 150 m depth-integrated $N_2$ fixation rate (INF) and primary production
(IPP) at each station. nd, not determined.

| Station | SST (°C) | SSS | Chl $a$ (μg/L) | $D_{Nitr}$ (m) | SNF (nmol N $L^{-1}$ $d^{-1}$) | INF (μmol N $m^{-2}$ $d^{-1}$) | IPP (mmol C $m^{-2}$ $d^{-1}$) |
|---|---|---|---|---|---|---|---|
| SEATS$_{2016}$ | 30.3 | 33.46 | 0.26 | 51 | 1.1 | 63 | 44 |
| SEATS$_{2018}$ | 30.3 | 33.46 | 0.11 | 71 | 1.8 | 123 | 24 |
| S1 | 29.5 | 33.73 | 0.24 | 27 | 0.8 | 54 | 43 |
| S2 | 29.4 | 33.75 | 0.10 | 50 | 3.0 | 103 | 24 |
| S3 | 28.7 | 33.53 | 0.15 | 30 | 2.4 | 181 | 98 |
| S4 | 29.5 | 33.74 | 0.17 | 62 | 1.8 | 247 | 59 |
| SK1 | 30.5 | 33.62 | 0.22 | 100 | 10.4 | 437 | 11 |
| SK2 | nd | nd | 0.11 | nd | 2.0 | nd | nd |
| K1 | 29.1 | 34.45 | 0.11 | 130 | 0.8 | 120 | 19 |
| WP | 30.9 | 34.47 | 0.11 | 163 | 1.9 | 403 | 9 |


A significant positive correlation was found between the depth-integrated *nifH* gene
abundance and $N_2$ fixation rate (Pearson $r = 0.72$, $p = 0.046$, Supplementary Table S2),
demonstrating that the abundances of these major diazotroph phylotypes well explained
the major variability in measured rates. However, considerable spatial variation was
found in the specific diazotrophs supporting $N_2$ fixation (Fig. 3). *Trichodesmium*





dominated the diazotroph assemblage throughout the water column of the NSCS (52-96%
of the total *nifH* gene abundance, excluding station SEATE$_{2016}$). In contrast, at the
Kuroshio station K1, unicellular diazotrophic cyanobacteria (UCYN-A and UCYN-B)
were the most abundant phylotypes, and at station WP, UCYN-B alone was dominant
(Fig. 3 and Supplementary Table S3).

**3.2 Diazotroph response to Fe and P supply**
To directly test which nutrients were limiting overall N$_2$ fixation rates and the abundance
of individual diazotrophs, we conducted eight, ~3-day nutrient addition bioassay
experiments (Figs. 4 and 5). The responses of N$_2$ fixation rate to different combinations
of Fe and P supply demonstrated a coherent geographic switch across the study area
(Figs. 1, 4 and 5). At stations towards to the NSCS basin (SEATS$_{2016}$, S1 and S3), N$_2$
fixation rates were co-limited by Fe and P. Two forms of this co-limitation were
identified: (i) only simultaneous Fe and P addition stimulated N$_2$ fixation rates
('simultaneous co-limitation', station S1, Fig. 4); (ii) independent addition of either Fe or
P alone, or supply of Fe and P in combination, enhanced N$_2$ fixation rates ('independent
co-limitation', stations SEATS$_{2016}$ and S3, Fig 4). Further to the northeast, in contrast, N$_2$
fixation rates were only stimulated by nutrient combinations containing Fe at S4 and by
combinations containing P at K1 and WP, suggesting single limitation by Fe or P,
respectively, at these sites (Fig. 4). Although Fe addition also appeared to stimulate N$_2$



fixation rates at station K1, P was generally the major limiting nutrient at this station
taking into account the responses of both $N_2$ fixation rates and *nifH* gene abundance (see
below) (Figs. 4 and 5). At stations (SK1 and SK2) in the Luzon Strait, mean $N_2$ fixation
rates were highest in treatments containing P, but responses were not significantly greater
than the untreated controls, suggesting that both Fe and P availability were not limiting
$N_2$ fixation rates (Fig. 4).

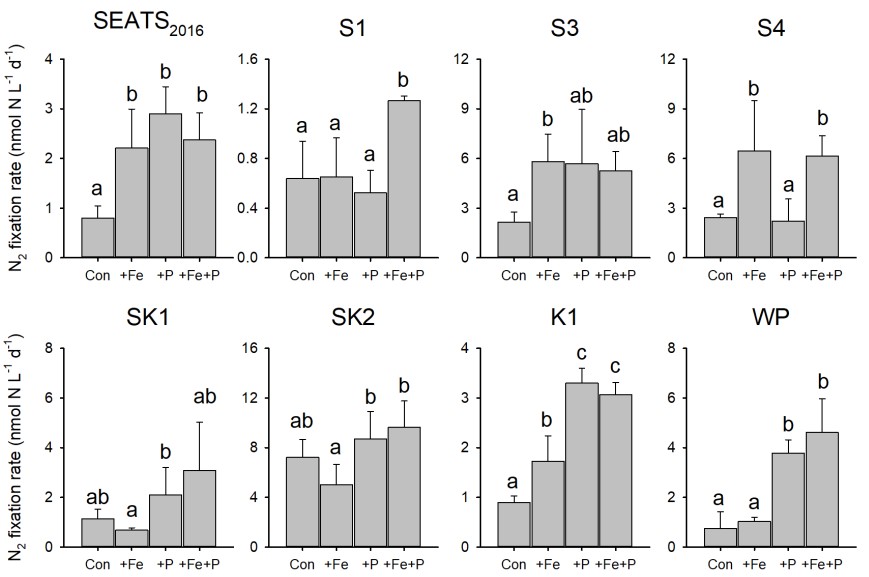

**Figure 4.** Response of $N_2$ fixation to nutrient amendment. Error bars represent the standard
deviation of biological replicates ($n = 2$ or $3$). Different letters above error bars indicate
statistically significant differences ($p < 0.05$) between treatments (ANOVA followed by
Fisher PLSD test).

Further detail as to the drivers of the $N_2$ fixation responses to Fe and P additions was

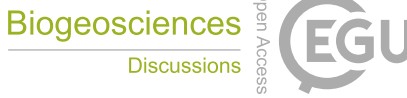

326 provided by the species-level analysis of diazotroph *nifH* from the treatment bottles. In

327 general, responses of total *nifH* gene abundance to Fe and P amendments were

328 qualitatively consistent with $N_2$ fixation rates at most sites, that is, the nutrient(s) limiting

329 $N_2$ fixation rates also limited the diazotroph abundance (Figs. 4 and 5). The exceptions

330 were at stations S3 and S4, where variability in *nifH* abundances was observed in

331 response to nutrient treatment (station S3) or overall trends differed between *nifH*

332 abundances and $N_2$ fixation rates (station S4; enhanced *nifH* abundance in response to +P,

333 whereas rates only responded to +Fe). Quantitatively, the responses of $N_2$ fixation

334

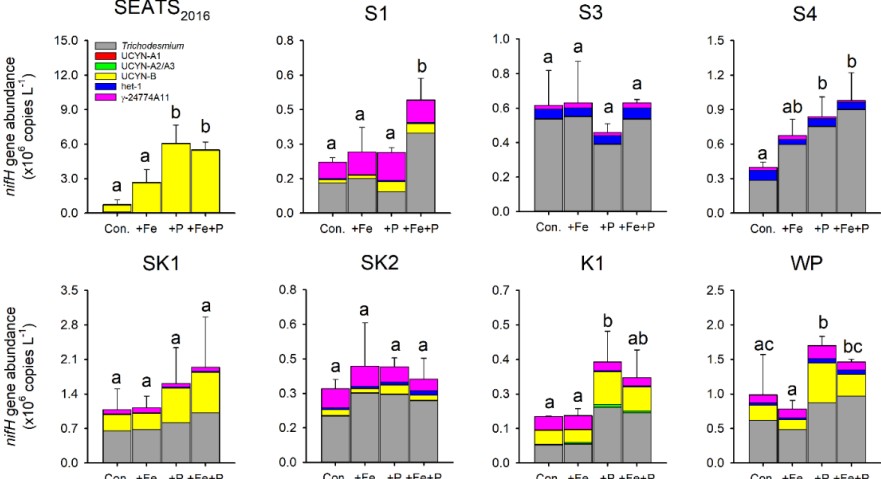

335

336

**Figure 5.** Response of diazotroph phylotypes to nutrient amendment. Bar heights
represent the mean total *nifH* concentration and error bars the standard deviation of
biological replicates ($n = 2$ or 3). Different letters above error bars indicate a statistically
significant difference ($p < 0.05$) between treatments (ANOVA followed by Fisher PLSD
test).




rates and *nifH* biomass to nutrient addition were not well correlated (total *nifH* abundance
increase rate versus $N_2$ fixation increase rate following nutrient supply, $R^2 = 0.07$, $p =$
0.21; Supplementary Fig. S1), despite initial background *nifH* abundances and $N_2$ fixation
rates being well correlated (Pearson $r = 0.72$, $p = 0.046$, Supplementary Table S1). This
suggested a decoupling of the rates of change in biomass and $N_2$ fixation rates following
nutrient addition over the relative short incubation timescales (~3 days).

Overall, the diazotroph community structure was not greatly changed after nutrient
amendments (Fig. 5). *Trichodesmium* and UCYN-B were the two most dominant species
in all experimental waters that contributed to the enhanced total *nifH* gene abundance
after nutrient additions (Figs. 3, 5 and Supplementary Fig. S1). Despite showing
independent co-limitation in response to Fe and P supply at station SEATS$_{2016}$ (Fig. 4), as
reflected by equally responding $N_2$ fixation rates, UCYN-B, the dominant diazotroph in
non-amended control waters, increased 2-fold more following P addition in comparison
to Fe addition (Fig. 5 and Supplementary Fig. S2). Furthermore, no significant changes in
*nifH* were observed at station S3, where $N_2$ fixation rates were also independently Fe-P
co-limited. More consistent between the $N_2$ fixation rates and *nifH* biomass changes were
the *nifH* responses at station S1, with overall *nifH* concentrations only responding to
Fe+P additions, matching the $N_2$ fixation response. This was mostly driven by co-
limitation of *Trichodesmium*, whereas UCYN-B responded only to P supply (Fig. 5 and



Supplementary Fig. S2).

In contrast to the Fe limitation of $N_2$ fixation rates found at station S4, *nifH* abundances
showed the most significant responses to the combined supply of Fe and P. However, at
sites where $N_2$ fixation rates were P-limited (K1 and WP) overall *nifH* concentrations also
responded most to P addition, with contributions from both *Trichodesmium* and UCYN-B
(Fig. 5). In addition, het-1 also increased significantly with +P combinations at stations
K1 and WP (Supplementary Fig. S2). By contrast, γ-24774A11, which also accounted for
a substantial fraction of the diazotroph community (up to 31%), did not show clear
enhancement to nutrient additions (Supplementary Fig. S2), suggesting that it was not Fe-
and/or P-limited.

**4 Discussion**
In the present study, rates and *nifH* gene abundances were much higher in the northeast
region of our study area than in the NSCS basin (Fig. 3). Rates at stations SK1 and WP
were comparable to those recently reported in this region (~450 µmol N $m^{-2}$ $d^{-1}$)
measured using the same $^{15}N_2$ gas dissolution method (Lu et al., 2019; Wen et al., 2022).
Although relatively low rates were measured at the Kuroshio Current station (K1)
compared with previous observations (e.g., Wen et al., 2022), high *nifH* gene abundance
was nevertheless observed at this site (Fig. 3 and Supplementary Table S3). Therefore,

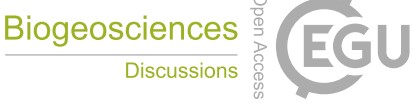

our observations provide increasing evidence for this western (sub)tropical North Pacific
boundary region containing important "hot spots" of $N_2$ fixation (Shiozaki et al., 2010;
Shiozaki et al., 2015a; Wen et al., 2022). However, the elevated total *nifH* concentration
in the western boundary of the North Pacific during our study was largely attributed to an
increased abundance of unicellular diazotrophs (UCYN-A and B, Fig. 3), but not
*Trichodesmium* as previously reported (Chen et al., 2003; Chen et al., 2014; Chen et al.,
2008; Shiozaki et al., 2014a). Instead, we found that *Trichodesmium* was most abundant
at stations (S3 and S4) close to the western edge of the Luzon Strait (Fig. 3 and
Supplementary Table S3), where Kuroshio intrusion water has been hypothesized to
introduce *Trichodesmium* into a favorable biogeographic regime (Lu et al., 2019). Either
this region is spatially and/or temporally heterogeneous with respect to the presence of
unicellular versus *Trichodesmium* diazotrophs, or the environmental changes have led to
a shift in diazotroph community structure (Gruber, 2011; Hutchins and Fu, 2017).

Depth-integrated $N_2$ fixation rate and *nifH* gene abundance were not correlated with sea
surface temperature (SST), but a significant positive correlation was found between
nitracline depth and total *nifH* gene abundance (Pearson $r = 0.74$, $p = 0.037$,
Supplementary Table S2). This was suggestive of subsurface N supply into the euphotic
zone, which is inversely related to nitracline depth, potentially being important in
regulating diazotroph abundance in our study area, with lower N supply leading to





enhanced diazotroph abundances (Chen et al., 2003; Shiozaki et al., 2014b). The presence
of diazotrophs in the ocean will be a function of how well they can compete with non-
diazotrophic phytoplankton for limiting resources (e.g., Fe and P) under grazing pressure
(Dutkiewicz et al., 2014; Landolfi et al., 2021; Ward et al., 2013). Accordingly, because
of the growth characteristics of diazotrophs in comparison to non-diazotrophs, in
particular their lack of requirement for pre-fixed N, but higher requirement for Fe and P,
the relative supply rates of N, Fe and P are highly important in dictating where
diazotrophs can succeed (Ward et al., 2013). Aligning with earlier global model
predictions (Ward et al., 2013), and investigations in the (sub)tropical Atlantic (Schlosser
et al., 2014), Wen et al. (2022) recently found that the Fe:N supply ratio (including
subsurface and aerosol N and Fe supplies) was a robust predictor of diazotroph standing
stock across the broader western North Pacific, including our study region.

Although the current study lacked the data to calculate nutrient supply rates into the
euphotic zone (matching Fe concentration profiles, euphotic depths), the correlation
found between *nifH* and nitracline depth suggested the potential for the same driver (i.e.,
Fe:N supply rates) to be operating over this smaller spatial scale. In line with Wen et al.
(2022), we further hypothesize that the expected significant N supply rate to surface
waters of the NSCS (due to a shallower nitracline, alongside riverine and aerosol inputs)
reduces, but does not eliminate the competitive ability of diazotrophs, as Fe supply rates



to this region are likely also high (Duce et al., 1991; Jickells et al., 2005; Zhang et al.,
2019), thereby maintaining Fe:N supply ratios at levels supporting diazotrophs (Ward et
al., 2013; Wen et al., 2022). At these Fe:N supply levels, we observed that $N_2$ fixation
rates were either (i) 'simultaneously co-limited' by Fe and P (identified at station S1),
which represents a state where two, non-substitutable nutrients (in this case, Fe and P)
have been drawn down to equally limiting levels (Sperfeld et al., 2016), or (ii)
'independently co-limited' (stations $SEATS_{2016}$ and S3), which represents a state where
the resources are substitutable at biogeochemical (Saito et al., 2008), or community levels
(Arrigo, 2005).

The measured contributions of individual diazotrophs to total *nifH* concentration in
response to nutrient supply suggested that simultaneous Fe-P co-limitation of $N_2$ fixation
rates at station S1 was via regulation of *Trichodesmium*, which only responded to Fe+P
addition (Fig. 5). The *nifH* responses also suggested that independent Fe-P co-limitation
of $N_2$ fixation rates at sites $SEATS_{2016}$ and S3 was not operating at the community level
(i.e., one diazotroph type limited by Fe and the other by P) (Arrigo, 2005), as different
diazotroph community structure responses to either Fe or P addition were not observed
(Fig. 5). We suggest three possible causes for this observation: (i) co-limitation was at the
biochemical rather than community level (i.e., either Fe or P could enhance the rates of
processes ultimately driving elevated $N_2$ fixation) (Saito et al., 2008); (ii) a more subtle





community co-limitation was occurring at the level of ecotypes not resolved by the *nifH*
qPCR analyses; or (iii) community co-limitation of $N_2$ fixation rates for the measured
groups was occurring, but, unlike the simultaneous co-limitation scenario at station S1,
experimental durations were too short for this to be reflected in diazotroph biomass
changes. Surprisingly, stations with independent co-limitation of $N_2$ fixation rates by Fe
and P (SEATS$_{2016}$ and S3) were not additive (i.e., increases in $N_2$ fixation rates in Fe+P
treatments were not larger than Fe and P alone) (Sperfeld et al., 2016). Although the
available data do not allow us to provide a concrete reason for this, it could reflect serial
limitation of $N_2$ fixation by another resource (e.g., a different nutrient or light).

In contrast to the more central NSCS, in the western boundary of the North Pacific,
elevated Fe:N supply ratios are expected as a result of deepening nitraclines (Fig. 2 and
Table 1) and continued aerosol Fe inputs (Wen et al., 2022). Additional Fe inputs other
than aerosol deposition may have also contributed to further enhanced Fe:N supply in the
Luzon Strait. At station SK2, much higher surface particulate Fe concentrations (both
intracellular and total forms) were observed (Supplementary Table S4), implying
additional Fe inputs, potentially sourced from the adjacent islands and the surrounding
shallow sub-surface bathymetry (Shiozaki et al., 2014a; Shiozaki et al., 2015a). In turn
we hypothesize that elevated Fe:N supply rates enhance $N_2$ fixation rates at these sites
(Fig. 3), which leads to P drawdown and subsequent P limitation of the enhanced

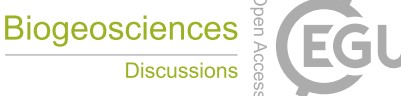

diazotroph stock (Figs. 4 and 5; Hashihama et al., 2009; Ward et al., 2013; Wen et al.,

464     2022).


In addition to the Fe:N supply ratio regulating the total *nifH* gene abundance and activity
(Wen et al., 2022), we also further hypothesize that overall Fe supply rates might be an
important factor in determining the diazotroph community structure in our study area
(Church et al., 2008; Langlois et al., 2008; Shiozaki et al., 2017). Specifically, the depth-
integrated diazotroph compositions switched from being co-dominated by *Trichodesmium*
and other diazotrophs in the central NSCS (SEATS, S1 and S2), *Trichodesmium*-
dominated in the more northern NSCS (S3 and S4), and finally dominated by UCYN-B in
the western boundary of the North Pacific (Fig. 3 and Supplementary Table S3). Elevated
Fe supply in the NSCS, particularly around the islands and shallow bathymetry of the
Luzon Strait, might create a more favorable condition for *Trichodesmium* (Fig. 3 and
Supplementary Table S3), consistent with elevated Fe demands of this species (Kupper et
al., 2008; Sohm et al., 2011), as well as its ability to use particulate Fe forms (Rubin et
al., 2011), and in line with the elevated contribution of this species found in other regions
with enhanced Fe supply (e.g., the tropical North Atlantic and western South Pacific;
Bonnet et al. 2018; Sañudo-Wilhelmy et al., 2001; Sohm et al., 2011; Stenegren et al.,
2018). Conversely, unicellular species may be more competitive than *Trichodesmium* in
regions with lower Fe supply rates (Fig. 3). In addition to having a higher surface to



volume ratio that favors Fe uptake (Hudson and Morel 1990; Jacq et al., 2014), UCYN-B
species such as *Crocosphaera* have been reported to employ a repertoire of Fe-
conservation strategies, e.g., daily synthesis and breakdown of metalloproteins to recycle
Fe between the photosynthetic and $N_2$ fixation metalloenzymes and increased expression
of flavodoxin at night even under Fe-replete conditions (Saito et al., 2011). These
potentially explain why UCYN-B was less Fe-limited in the NSCS basin (stations
SEATS$_{2016}$ and S1; Fig. 5 and Supplementary Fig. S2) and dominates the diazotroph
community on the western Pacific side of the Luzon Strait (Fig. 3; Chen et al., 2019; Wen
et al., 2022).

**5 Conclusions**
Observations and experiments conducted in the NSCS and the western boundary of the
North Pacific demonstrated that in the more central NSCS, Fe and P were co-limiting the
lower overall observed $N_2$ fixation rates, whereas P was limiting the higher rates on the
western Pacific side of the Luzon Strait. This matched the expectation of higher Fe:N
supply ratios in the western Pacific generating a more favorable niche for diazotrophs,
leading to a drawdown of P. *Trichodesmium* and UCYN-B were the most dominant
diazotroph types in the incubation waters and both dominated the responses of the total
*nifH* gene after nutrient amendments. In general, nutrient addition had a relatively
restricted impact on diazotroph community structure apart from on UCYN-B, which





showed increased contribution in the diazotroph community following P addition at sites
where $N_2$ fixation rates were P-limited. We hypothesize that overall switches in
diazotroph community structure from *Trichodesmium*-dominated in the NSCS to single-
celled UCYNA/B was related to declines in overall Fe supply rates and the different
physiological strategies of these diazotrophs to obtain and use Fe. Future research that
more accurately constrains nutrient supply rates to these different regions would be
beneficial for further resolving this hypothesis.



*Data availability.* All data needed to evaluate the conclusions in the paper are present in
the paper and/or the Supplementary Materials. Additional data associated with the paper
are available from the corresponding authors upon request.

*Author contributions.* D.S., H.H., and Z.W. designed the research. Z.W., R.D., W.W.,
W.L., X.H., W.L., and L.W. performed the experiments. Z.W., D.S., H.H., T.J.B., X.L.,
and Z.C. analyzed the data. Z.W., T.J.B., H.H., and D.S. wrote the manuscript. All authors
discussed the results and commented and edited the manuscript.

*Competing interests.* The authors declare that they have no conflict of interest.

*Acknowledgements.* The authors acknowledge the captains and crew of the R/V
*Dongfanghong* 2 and R/V *Tan Kah Kee* for the help during the cruises. This work was
supported by the National Science Foundation of China (41890802, 42076149,
41925026, 42106041, and 41721005), the "111" Project (BP0719030), and the
XPLORER Prize from the Tencent Foundation to D. Shi.



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
