# Peer review of "The response of diazotrophs to nutrient amendment in the"

_Biogeosciences, 2022_

## Author Comment (AC1)

**Reviewer 1**

General comments
In their manuscript bg-2022-155, Wen et al. present the results of a series of nutrient additions experiments conducted in the South China Sea and western North Pacific, where in addition to bulk $N_2$ fixation rates, species composition based on *nifH* gene abundance was analyzed in response to Fe and P amendments.

Overall, I enjoyed reading the manuscript, and I find the dataset is a useful addition to our understanding of the regulation of $N_2$ fixation rates and how it is linked to species composition. The manuscript is well written, and the discussion insightful. Yet, there are a few points that I believe should be discussed/improved, specifically I have some concerns/queries with regard to replication and interpretation of qPCR results, as outlined below.

We thank the Reviewer for the positive evaluation of our manuscript, and provide responses to specific comments below.

Specific comments
Apparently, in several of the experiments there were only 2 replicates – I wonder whether the statistical analysis procedures are valid for two replicates? Could the authors at least indicate in each of the figures which data are averages of 2?

Treatments for most of the bioassay experiments (7 out of 8) were conducted with 3 replicates. However, there were three cases when one of the triplicate samples was lost due to filtration errors (i.e., one +Fe+P carboy at station S1, one NFR/PP sample of +Fe+P at station WP, and one +P sample at station S3). In addition, for the bioassay experiment at station SEATS_2016, sufficient water was only available to conduct the experiment with 2 replicates for control and +Fe+P treatments, while +Fe and +P groups retained 3 replicates. Further details outlining the above information have now been added to the Methods section and also in the figure legends of Figures 4 and 5.

The authors acknowledge that $^{15}N$ label% was not measured, which is indeed a shortcoming, but as they state that the experimental procedure and the results were comparable to their previous study, I believe it is acceptable. An average and stdev of label% in the previous study are given, but could the authors add any further details to help us understand how reproducible the approach was (number of replicates etc)?

We sampled the $^{15}N_2$ atom% for $N_2$ fixation measurements using exactly the same approach during a cruise in 2020. The $^{15}N_2$ atom% values in the incubation bottles from the cruise are shown in table below. Only one replicate was measured for the $^{15}N_2$ atom% for each of the station depths. However, the relatively narrow range of 1.28% to 1.56% (mean ± s.d. of 1.40 ± 0.08, n = 17) suggests that the preparation, sampling, and measurement of $^{15}N_2$ were stable and reproducible.

Measured $^{15}N_2$ atom% in incubation bottles in a subsequent cruise to the western North Pacific during winter 2020.

| Station | Longitude (degrees_east) | Latitude (degrees_north) | Depth (m) | $A^{15}N_2$ (atom %) |
|---|---|---|---|---|
| K11 | 118.5 | 21.5 | 5 | 1.37 |
| K11 | 118.5 | 21.5 | 15 | 1.41 |
| K11 | 118.5 | 21.5 | 25 | 1.43 |
| K11 | 118.5 | 21.5 | 40 | 1.33 |
| K11 | 118.5 | 21.5 | 90 | 1.41 |
| K11 | 118.5 | 21.5 | 140 | 1.42 |
| K11<10 µm | 118.5 | 21.5 | 5 | 1.53 |
| K11<10 µm | 118.5 | 21.5 | 25 | 1.52 |
| K11<10 µm | 118.5 | 21.5 | 140 | 1.56 |
| K8a | 155.0 | 12.5 | 5 | 1.28 |
| K8a | 155.0 | 12.5 | 25 | 1.31 |
| K13a | 131.0 | 11.0 | 5 | 1.41 |
| K13a | 131.0 | 11.0 | 150 | 1.35 |
| UW-127 | 137.6 | 13.4 | Surface | 1.39 |
| ZH-56 | 126.0 | 20.2 | Surface | 1.39 |
| ZH-58 | 124.0 | 20.4 | Surface | 1.35 |
| ZH-61 | 121.0 | 21.0 | Surface | 1.33 |

I wonder how the reports on *Trichodesmium* polyploidy (ca 100 genome copies per cell in field samples, e.g. Sargent et al. 2016 https://doi.org/10.1093/femsle/fnw244) affect the estimates of species composition based on *nifH* gene copies, as well as the trends observed in bioassays. Was polyploidy taken into account when the 'dominant species (e.g., l. 351)' were determined? And, taking into account the high level of polyploidy in *Trichodesmium* compared to the other species, couldn't shifts in the species composition explain the mismatch in responses of $N_2$ fixation vs *nifH* abundance (e.g., l. 343)?

We agree with the Reviewer that polyploidy may have an important impact on the estimates of diazotroph compositions. Given that the degree of polyploidy can vary significantly (ranging from 1 to 1405; Sargent et al., 2016; White et al., 2018), with a potential dependence on the growth conditions, nutrient status, developmental stage, and cell cycle (see references in Karlusich et al., 2021), we do not attempt to account/correct for this in calculations of proportions of the different diazotrophs. We however have made revisions to the text to stress that polyploidy might impact the assumed proportions of the different diazotrophs under both in situ and nutrient amended conditions (with associated references).

Related to this, I would suggest being more cautious about the use of the terms 'abundance' (e.g. abstract l. 26 'abundances of specific diazotrophs', l. 119 'abundances of specific diazotroph phyla') and 'growth' where actually gene copy number was measured. Specifically, in supplementary figure 1, I would suggest replacing 'growth rate' by 'gene abundance', since growth rate might imply that measurements of cell density or C concentration were made.

Thanks for this suggestion. Changes have now been made to the manuscript. For example, we emphasize it was *nifH* gene abundance that we measured. Also, the "growth rate" in Figure S1 has been replaced by "increase rate".

Minor comments
34-35 'the largest responses were always dominated by either *Trichodesmium* or UCYN-B': this is not clear (responses in what?) – can it be clarified?

This has now been clarified as "The largest responses in *nifH* gene abundances…".

110-112 it is not completely clear from this why high spatial resolution is necessary - can this be justified better?

The Kuroshio intrusion generates a frontal zone with a unique diazotrophy regime in the NSCS (Lu et al., 2019). Therefore, the lower spatial resolution of the experiments in Wen et al. (2022) and other studies (Chen et al., 2019; Shiozaki et al., 2014b) remains insufficient to delineate Fe and P controls on diazotrophy at finer spatial scales between the neighboring NSCS and the western boundary of the North Pacific. This justification has now been further refined in the revised manuscript.

164 can more details on the gas-tight plastic bags (supplier) be added?

Details added: "Tedlar®PVF, Dalian Delin Gas Packing Co., Ltd".

221 why were those Fe and P concentrations chosen – are there any references to add on how these relate to in situ concentrations in this area?

Surface dissolved Fe and P concentrations previously reported in the NSCS were ~0.17-1.01 nM and ~5-20 nM respectively (Wu et al., 2003; Zhang et al., 2019). In order to obtain a measurable response within the relatively short 72-h experimental period, approximately 5 times higher concentrations were added into the incubation bottles. Details have now been added to the Methods.

223 Can the authors supply some more details on the incubation system? Do I understand correctly that 10L carboys were placed in the on-deck incubator, and there was some kind of water jacket flushed with seawater for temperature control?

The 10 L carboys were placed in a ~400-L clear on-deck incubator with inflow and outflow. Surface seawaters were then pumped into the incubator for temperature control. Details have now been added to the revised Methods.

Please supply more detail or a reference on the method for chl measurement.

More details and reference (Welschmeyer 1994) have now been added in the Methods.

441 I believe the biochemical substitution of Fe and P deserves some more explanation (either here or at a later stage) - how could this work, are there specific mechanism/enzymes that can substitute Fe for P?

We suggested that this substitution was not directly between Fe and P, but indirectly between the resources or the abilities of resource acquisition controlled by Fe and P. For instance, in addition to serving as cofactors of nitrogenase, Fe is also cofactors of alkaline phosphatases (Rodriguez et al., 2014; Yong et al., 2014). Thus, the addition of Fe may enhance the utilization of dissolved organic P (DOP) in the face of low level dissolved inorganic P (DIP) (Browning et al., 2017). Explanations have now been added to the revised text.

450-451 also the 'serial limitation' of $N_2$ fixation by another resource deserves a few more words for explanation I believe – it is not clear how this would work

A serial limitation, also termed secondary limitation, is the scenario where only the addition of one resource shows a positive response (either resource 1 or 2) and the addition of both resources together shows a bigger response than the primary, single limiting resource. This means, for example, the addition of resource 1 has led to growth which leads to depletion of resource 2 in the system. Thus, further addition of resource 2 leads a secondary growth response. In our case, $N_2$ fixation at stations SEATS2016 and S3 were independently co-limited by Fe and P, and the addition of Fe and P simultaneously should have led an additive rate response of $N_2$ fixation rates (Sperfeld et al., 2016). However, the absent of this additive response in our study may reflects that either addition of Fe or P has depleted other secondary limitation nutrients (e.g., Ni), or that the overall light level during our incubation has set an upper limit of $N_2$ fixation rate, which both have prevented further rates enhancement after nutrient additions.

476 I think there might be more specific refences for the Fe demand of *Trichodesmium* (e.g., Kustka et al., https://doi.org/10.1016/S0923-2508(02)01325-6, 10.4319/lo.2003.48.5.1869, https://doi.org/10.1046/j.1529-8817.2003.01156.x)

We thank the Reviewer for providing this information. This reference has now been added to the revised manuscript.

Could it be specified how the data in Fig. S1 was calculated? Is the growth rate in units d-1 the total increase in gene abundance over the 3 days bioassay experiment divided by 3?

The relative growth rates (now change to increase) were defined as the relatively changes of $N_2$ fixation rates or *nifH* abundances after nutrient additions compared to the control. The equation for the increase rate calculation is Ln (*nifH*_treatment / *nifH*_control) / time, where time is in days. Details were added in the main text and figure legend of Figure S1.

In the supplementary table showing parameters involved in nitrogen fixation rate calculations, the units for several of the parameters (e.g., depth, chl a, PON, NFR) are missing in the headers, please add these.

Units have been added in the supplementary table.

Technical corrections
62-63 check the order of references: in sequence of publishing year?

Corrected.

69 *did* not quantitatively match

Corrected.

72 *is* potentially crucial

Corrected.

76 I assume this should mean 'contribute differently to the sinking flux carbon that small unicellular species (i.e., delete 'that of')'?

Corrected.

110 remain*s*

Corrected.

242 metal bound (remove hyphen)

Corrected.

266 below

Corrected.

313 and others: remove brackets around station names in the text

Corrected.

References:

Browning, T. J., Achterberg, E. P., Yong, J. C., Rapp, I., Utermann, C., Engel, A., and Moore, C. M.: Iron limitation of microbial phosphorus acquisition in the tropical North Atlantic, 8, 15465, 10.1038/ncomms15465, 2017.

Chen, M., Lu, Y., Jiao, N., Tian, J., Kao, S. J., and Zhang, Y.: Biogeographic drivers of diazotrophs in the western Pacific Ocean, Limnol. Oceanogr., 64, 1403-1421, 10.1002/lno.11123, 2019.

Karlusich, J. J. P., Pelletier, E., Lombard, F., Carsique, M., Dvorak, E., Colin, S., Picheral, M., Cornejo-Castillo, F. M., Acinas, S. G., Pepperkok, R., Karsenti, E., de Vargas, C., Wincker, P., Bowler, C., Foster, R. A.: Global distribution patterns of marine nitrogen-fixers by imaging and molecular methods, Nature Communications, 12, 10.1038/s41467-021-24299-y, 2021.

Lu, Y., Wen, Z., Shi, D., Lin, W., Bonnet, S., Dai, M., and Kao, S. J.: Biogeography of $N_2$ fixation influenced by the western boundary current intrusion in the South China Sea, J Geophys Res-Oceans, 124, 6983-6996, 10.1029/2018jc014781, 2019.

Rodriguez, F., Lillington, J., Johnson, S., Timmel, C. R., Lea, S. M., and Berks, B. C.: Crystal structure of the bacillus subtilis phosphodiesterase PhoD reveals an iron and calcium-containing active site, J. Biol. Chem., 289, 30889-30899, 10.1074/jbc.M114.604892, 2014.

Sargent, E. C., Hitchcock, A., Johansson, S. A., Langlois, R., Moore, C. M., LaRoche, J., Poulton, A. J., and Bibby, T. S.: Evidence for polyploidy in the globally important diazotroph *Trichodesmium*, FEMS Microbiol. Lett., 363, 10.1093/femsle/fnw244, 2016.

Shiozaki, T., Chen, Y. L. L., Lin, Y. H., Taniuchi, Y., Sheu, D. S., Furuya, K., and Chen, H. Y.: Seasonal variations of unicellular diazotroph groups A and B, and *Trichodesmium* in the northern South China Sea and neighboring upstream Kuroshio Current, Cont. Shelf Res., 80, 20-31, 10.1016/j.csr.2014.02.015, 2014.

Sperfeld, E., Raubenheimer, D., and Wacker, A.: Bridging factorial and gradient concepts of resource co-limitation: Towards a general framework applied to consumers, Ecol. Lett., 19, 201-215, 10.1111/ele.12554, 2016.

Welschmeyer, N. A.: Fluorometric analysis of chlorophyll-a in the presence of chlorophyll-B and pheopigments, Limnol. Oceanogr., 39, 1985-1992, 1994.

Wen, Z., Browning, T. J., Cai, Y., Dai, R., Zhang, R., Du, C., Jiang, R., Lin, W., Liu, X., Cao, Z., Hong, H., Dai, M., and Shi, D.: Nutrient regulation of biological nitrogen fixation across the tropical western North Pacific, Science Advances, 8, eabl7564, 10.1126/sciadv.abl7564, 2022.

White, A. E., Watkins-Brandt, K.S., and Church, M. J.: Temporal variability of *Trichodesmium* spp. and diatom-diazotroph assemblages in the North Pacific Subtropical Gyre, Frontiers in Marine Science, 5, 10.3389/fmars.2018.00027, 2018.

Wu, J., Chung, S. W., Wen, L. S., Liu, K. K., Chen, Y. L. L., Chen, H. Y., and Karl, D. M.: Dissolved inorganic phosphorus, dissolved iron, and *Trichodesmium* in the oligotrophic South China Sea, Global Biogeochem. Cycles, 17, 1008, 10.1029/2002gb001924, 2003.

Yong, S. C., Roversi, P., Lillington, J., Rodriguez, F., Krehenbrink, M., Zeldin, O. B., Garman, E. F., Lea, S. M., and Berks, B. C.:  A complex iron-calcium cofactor catalyzing phosphotransfer chemistry, 345, 1170-1173, 10.1126/science.1254237, 2014.

Zhang, R., Zhu, X., Yang, C., Ye, L., Zhang, G., Ren, J. L., Wu, Y., Liu, S. M., Zhang, J., and Zhou, M.: Distribution of dissolved iron in the Pearl River (Zhujiang) Estuary and the northern continental slope of the South China Sea, Deep Sea Research Part II, 167, 14-24, 10.1016/j.dsr2.2018.12.006, 2019.

---

## Author Comment (AC2)

**Reviewer 2**

In this paper, Wen et al. reported the patterns and limiting factors of nitrogen fixation activity and diazotroph community in the South China Sea and western North Pacific. Through the iron and phosphate amendment experiments, they delineated the nutrient limitations of nitrogen fixation and diazotrophs for different water bodies in the study region. These findings may facilitate our understanding of the bottom-up controls of the diazotrophs in the study region, while I have some concerns regarding the methods, interpretations, and discussions in the current manuscript.

This manuscript gives me a general impression that the authors kept mentioning another paper of their group (Wen et al., 2022), instead of sticking to the findings of the current study. For example, they discussed a lot about the iron to nitrogen supply ratio, which seems like a highlight in Wen et al., 2022, but this ratio was not analyzed in the current study. Also, the dissolved iron to dissolved inorganic nitrogen ratio was not available in this manuscript. I suggest the authors focus more on the direct and interactive effects of iron and phosphorus availability on the diazotrophs for the discussion.

We thank the Reviewer for this suggestion., and fully agree with the Reviewer that although it appears that the iron to nitrogen supply ratio was important in regulating diazotroph biogeography in this study, this ratio was not directly analyzed and thus should not be overly highlighted. We have now reduced discussion on Fe:N supply ratio and shifted the focus further to the diazotroph nutrient limitations and their potential causes.

Given that the iron and phosphorus limitations of diazotrophs are the major focus of this study, the authors should describe the iron and phosphorus availability in the study region based on their real data, like ambient concentrations of iron and phosphate, instead of speculating. Also, the iron and phosphate concentrations in the nutrient amendment experiments should also be reported. This information is particularly important when the authors discuss the reasons and biogeochemical implications for the experimental results.

Surface low-level inorganic nutrients have now been added in Table 1 and also discussed in the text. Unfortunately, in this study, iron concentrations in surface seawater were not measured, nor were nutrient concentrations within the individual incubation bottle. However, the experimental results we have obtained in the present study can be explained very well by using previously reported nutrient concentrations in the similar region.

Besides, the authors should be aware that they only analyzed some commonly observed diazotroph groups (i.e., part of the diazotroph community) with qPCR assays. In other words, some of the unanalyzed diazotrophs might be important N-fixers at some stations. Some unanalyzed diazotrophs might even pop up during the 3-day incubation. So, they need to be cautious when comparing the patterns of nitrogen fixation rates and diazotroph abundances. The authors may also consider conducting *nifH* amplicon sequencing for reconstructing the whole diazotroph community in the study region. Also, initial diazotroph abundances of the incubations should be reported as well.

We fully agree with the Reviewer that our qPCR-based analysis of *nifH* community may neglect some other diazotrophs. We have now revised the text to note this caveat (alongside the polyploidy comment made by Reviewer 1) when discussing diazotroph abundances and community structure.

However, we also note that an exhaustive analysis of diazotroph community structure using high-throughput sequencing has been done in the similar region of our study (Ding et al., 2021). The results showed that *Trichodesmium*, UCYN-A and B, and γ-24774A11 were indeed the main species that contributed >80% of the diazotroph community. Thus, we believe that our qPCR analysis has nevertheless captured the main diazotroph phylotypes that commonly exist in the NSCS and WNP.

Unfortunately, the initial diazotroph abundances at the beginning (i.e., t=0) of the nutrient amendment experiments were not available. The *nifH* abundances of the seawater from the CTD deployed at exactly the same locations as the nutrient amendment experiments were reported, which we assume represented the initial conditions.

L28-30: It is better to avoid hypothesis/speculation in the abstract. The iron to nitrogen supply ratio was not directly measured in this study as the authors stated in L416.

Thanks for the suggestion. The hypothesis has been removed.

L34-L35: "the largest" and "always" seem subjective and inaccurate based on Figure 5. Also, there was no significant response at some stations where *Trichodesmium* dominated.

The sentence has been rephrased as "the largest responses of *nifH* gene abundances were dominated by either *Trichodesmium* or UCYN-B in 6 out of 8 experiments".

L38-40: Why? I did not see any evidence from this study supporting this speculation directly.

We realized that the hypothesis that we put forward was speculative, so the sentence has been changed to "This study provides comprehensive evidence of nutrient controls on diazotroph biogeography in the margin of western North Pacific Ocean."

L136-137: This sentence is not informative, as the depths are not labeled in the figure. Nevertheless, the depths of the seafloor are not important here.

Sentence deleted.

L152-156: Were the water samples collected from different depths (2-5m) exposed to different degrees of dissolved iron contamination from the research vessel? Did the authors measure dissolved iron concentration for these water samples?

The stated 2-5 m depth range refers to the range that the tow-fish moved continuously during sailing (i.e., with movement of the vessel, passage of waves etc.).

Dissolved Fe concentrations were not measured during these cruises. However, in a cruise during summer 2019, surface waters were sampled using the same method for the measurements of dFe, and the results showed values of 0.43 nM at a near-shelf station and 0.27 nM at the basin station SEATS (Wen et al., 2022). These concentrations were comparable to 0.2-0.3 nM previously reported in the NSCS basin (Wu et al., 2003). Thus, we believe our sampling approach meets trace metal clean standards, and that the water samples were not contaminated by the research vessels.

L160: The results of primary production were not described or discussed. Was primary production also measured in the nutrient amendment experiment? The Chl-a and primary production from the experiments may be helpful when the authors discuss/speculate about the competition between diazotroph and non-diazotrophic phytoplankton (L28; L403).

Chl *a* from the experiments were measured and the results and discussions have now been added in the supplementary material Fig. S3 and the main text. Briefly, Chl *a* concentrations were not significantly affected by the amendments of nutrients, which in combination with the low concentrations of surface DIN implies that the overall phytoplankton community in both NSCS and western boundary of North Pacific was N-limited.

L226: Did you measure iron concentration in the $^{15}N_2$ enriched water? The preparation of $^{15}N_2$ enriched water may introduce iron contaminants (Klawonn et al., 2015).
Klawonn, I., Lavik, G., Böning, P., Marchant, H. K., Dekaezemacker, J., Mohr, W., & Ploug, H. (2015). Simple approach for the preparation of $^{15}N_2$-enriched water for nitrogen fixation assessments: evaluation, application and recommendations. Frontiers in microbiology, 6, 769.

We did not measure iron concentration in the enriched water. We note that all the materials including the degas unit and Tedlar®PVF bag coming in contact with the $^{15}N_2$ enriched water were acid-washed in a Class-100 cleanroom before use. We therefore believe restricted iron contaminants were introduced into the enriched water. In addition, we observed enhancement of $N_2$ fixation rates after Fe addition in this study, suggesting that any contaminating Fe (if there was) was not enough to stimulate $N_2$ fixation. This was also found in in a previously study in which the same approach was used (Wen et al., 2022).

L307: The description of S3 is confusing. Does "ab" in Figure 4 mean no significant difference with a and b? If that's the case, S3 seems not "independent co-limited". Please clarify.

Yes, rates in +P and +Fe+P were not significantly different from control. However, the averaged rates were increased by 1.64 and 1.44 times relative to control, which were comparable to the degree of enhancement in +Fe (1.70 times). Thus, $N_2$ fixation at this station can also cautiously be recognized as independently limited. We have carefully revised the description in the revised manuscript to: "In the experiments conducted at station S3, $N_2$ fixation was also recognized to be independent co-limited, the rates in all nutrient-amended groups increased by 1.44-1.70 times compared to control, although statistical significances were not observed in +P and +Fe+P (Fig. 4)."

L317-323: Please clarify the exact number of replicates for each treatment. Also, I doubt the statistical significance based on duplicates (n=2). The limitation of replication should be stated clearly in the manuscript. It is also the same for Figure 5. Were initial nitrogen fixation rates (i.e., the rates of the seawater from the pump) measured?

Treatments for most of the bioassay experiments (7 out of 8) were conducted with 3 replicates. However, there were three cases when one of the triplicate samples was lost due to filtration errors (e.g., one +Fe+P carboy at station S1, one NFR/PP sample of +Fe+P at station WP, and one +P sample at station S3). In addition, for the bioassay at station SEATS_2016, sufficient water was only available to conduct the experiment with 2 replicates for control and +Fe+P treatments, while +Fe and +P groups retained 3 replicates. Further details outlining the above have now been added to the Methods section and also in the figure legends of Figures 4 and 5.

Unfortunately, the initial $N_2$ fixation rates of the seawater from the pump were not measured in this study. The rates we measured (with seawater from the CTD) were at exactly the same locations as where the bioassay experiments were set up, which we assume can represent the initial rates.

L350: As you only analyzed part of the diazotroph community, you may consider revising "diazotroph community structure" to "abundances of analyzed diazotrophs".

Revised accordingly.

L367 and Figure 5: How about UCYN-A1? UCYN-A1 was also abundant at K1 and WP based on Figure 3, while they disappeared in the nutrient amendment experiment (Figure 5). Also, the initial diazotroph abundances should also be displayed in Figure 5.

Unfortunately, the initial diazotroph abundances at the very beginning of the nutrient amendment experiments were not available. We measured he *nifH* abundances of the seawater from the CTD deployed at exactly the same locations as where the bioassay experiments were setup, which we assume can represent the initial condition. The shift of diazotroph composition in the bioassay incubations at K1 and WP could be attributed to the three-days incubation times and thus "bottle effects" (Göran et al., 2003). More discussions have been included in the revised manuscript.

L382: There is no doubt about Kuroshio being a hotspot of nitrogen fixation, while the low rate at K1 is not the "increasing evidence" as stated here.

We thank the Reviewer for this comments. We think that the "hot spots of $N_2$ fixation" in the Kuroshio not only represents high $N_2$ fixation rates but also the abundant diazotrophs in this region. In our study, higher $N_2$ fixation rate was not found at station K1, but much higher diazotroph biomass were observed here compared to that in the NSCS basin. We agree with the Reviewer (see the comment below) that "Abundance of diazotrophs do not necessarily mean their contribution to nitrogen fixation". However, the abundant diazotrophs here may imply a potential of high $N_2$ fixation activities supported by the favorable environmental conditions of this "hot spot". The coincidental lower rate we observed at K1 could have been caused by the

environmental conditions during our investigation, which do not necessarily mean the loss of high $N_2$ fixation potential, given the abundant diazotrophs we observed.

L385: Abundances of diazotrophs do not necessarily mean their contribution to nitrogen fixation.

Please see the response above.

L412-431, 453-464, etc.: The contents (mostly iron to nitrogen supply ratio) of Wen et al. 2022 are worth mentioning, but, they should be reduced significantly in the discussion. As said, the iron to nitrogen supply ratio was not analyzed in this study.

We thank for the Reviewer for this suggestion. As mentioned, although it appears that the iron to nitrogen supply ratio was important in regulating diazotroph biogeography in this study, this ratio was not directly analyzed and thus should not be overly highlighted here. We have now reduced discussion on Fe:N supply ratio and shifted the focus further to the diazotroph nutrient limitations and their potential causes.

L440: I think another reason would be that the analyzed groups did not represent the entire diazotroph community. There could be other diazotroph groups, which were not analyzed in this study, influenced by treatments.

We agree. A discussion of this possibility has been added as follow. "Other diazotrophs which were not analyzed by the qPCR assay, may be responsible for the enhanced $N_2$ fixation rates after nutrient additions"

L473-475: However, the nitrogen fixation at S3 and S4 was mostly iron-limited, while the *Trichodesmium* abundances there were not affected by iron addition treatment. All these pieces of finding should be considered when you discuss the relationship between iron and *Trichodesmium* in the NSCS.

We agree with this comment. In fact, significant enhancement of *Trichodesmium* abundance after Fe addition was observed in the experiment conducted at station S4 (see Fig. S2). However, at station S3, Fe-stimulation effect was only observed with $N_2$ fixation rate but not *Trichodesmium* abundance. This probably reflects a decouple of $N_2$ fixation rate and diazotroph abundance under specific environmental conditions. Nevertheless, regulation of Fe supply on diazotroph community structure remains a hypothesis that is difficult to test directly with the available experimental data.

References:

Ding, C., Wu, C., Li, L., Pujar, L., Zhang, G., and Sun, J..: Comparison of diazotrophic composition and distribution in the South China Sea and the Western Pacific Ocean, Biology, 10, 10.3390/biology, 2021.

Göran, E. and Cooper, S. D.: Scale effects and extrapolation in ecological experiments, Adv. Ecol. Res., 33, 161-213, 10.1016/S0065-2504(03)33011-9, 2003.

Wen, Z., Browning, T. J., Cai, Y., Dai, R., Zhang, R., Du, C., Jiang, R., Lin, W., Liu, X., Cao, Z., Hong, H., Dai, M., and Shi, D.: Nutrient regulation of biological nitrogen fixation across the tropical western North Pacific, Science Advances, 8, eabl7564, 10.1126/sciadv.abl7564, 2022.

Wu, J., Chung, S. W., Wen, L. S., Liu, K. K., Chen, Y. L. L., Chen, H. Y., and Karl, D. M.: Dissolved inorganic phosphorus, dissolved iron, and *Trichodesmium* in the oligotrophic South China Sea, Global Biogeochem. Cycles, 17, 1008, 10.1029/2002gb001924, 2003.